# *Helraiser* intermediates provide insight into the mechanism of eukaryotic replicative transposition

Ivana Grabundzija[1], Alison B. Hickman[1] & Fred Dyda[1]

*Helitrons* are eukaryotic DNA transposons that have profoundly affected genome variability via capture and mobilization of host genomic sequences. Defining their mode of action is therefore important for understanding how genome landscapes evolve. Sequence similarities with certain prokaryotic mobile elements suggest a "rolling circle" mode of transposition, involving only a single transposon strand. Using the reconstituted *Helraiser* transposon to study *Helitron* transposition in cells and in vitro, we show that the donor site must be double-stranded and that single-stranded donors will not suffice. Nevertheless, replication and integration assays demonstrate the use of only one of the transposon donor strands. Furthermore, repeated reuse of *Helraiser* donor sites occurs following DNA synthesis. In cells, circular double-stranded intermediates that serve as transposon donors are generated and replicated by *Helraiser* transposase. Cell-free experiments demonstrate strand-specific cleavage and strand transfer, supporting observations made in cells.

[1] Laboratory of Molecular Biology, National Institute of Diabetes and Digestive and Kidney Diseases, National Institutes of Health, Bldg. 5, Bethesda, MD 20892-0560, USA. Correspondence and requests for materials should be addressed to F.D. (email: Fred.Dyda@nih.gov)

Transposable elements (TEs) are discrete pieces of DNA that possess the ability to move within their host genome via a process called transposition. Although initially deemed "junk DNA," they have since been found to be integral components of genome evolution, organization, and stability. Major evolutionary innovations, such as the vertebrate adaptive immune system[1,2] and placenta[3,4], as well as novel transcription factors and regulatory networks (reviewed in refs. [5,6]), have their origin in transposition, and active mobile elements continue to reshape genomes of prokaryotes and eukaryotes alike. This genome-transforming property can impact both the evolutionary trajectory of a transposon host (reviewed in ref. [7]) and its well-being, through the potential of TEs to induce mutations leading to disease or oncogenesis[8,9].

DNA transposons have been turned into valuable genetic tools[10,11] and possibilities for their application range from functional genomics, genome editing, and transgenesis to gene therapy (reviewed in ref. [12]). Remarkably, most eukaryotic DNA transposons and all those that have found applications to date are the so-called "cut-and-paste" type DNA transposons. These precisely cut themselves out of the donor DNA site by introducing double-strand breaks, and integrate the mobilized DNA segment to a target site, often with little or no sequence preference. The original copy of the transposon is lost at the donor site. The needed endonucleolytic and transesterification activities are always catalyzed by a variant of a RNase H-like catalytic domain[13].

*Helitrons*, a unique group of eukaryotic DNA transposons that have generated unusually extensive genome variation, were discovered in 2001[14]. They are found throughout the eukaryotic kingdom, sometimes comprising a significant portion of their host's genome (reviewed in refs. [15,16]). Due to their ability to capture and mobilize host genomic fragments, *Helitrons* have been shown to disseminate genomic regulatory elements[17,18], generate gene fragment duplications[19], and chimeric transcripts[18,19], create putative microRNA-binding sites[18], as well as

to impact the functional organization of their host's gene regulatory networks[20] and influence intraspecies diversity[19].

Our understanding of the mechanism of *Helitron* transposition is limited. Until recently, the characteristics of *Helitron* transposition were inferred by relying on in silico analysis as no currently active *Helitrons* have been identified, preventing direct experimental work. Still, it is clear that they are distinct from any of the other characterized eukaryotic DNA transposons (reviewed in refs. [15,16]). Unlike "cut-and-paste" transposons, they do not contain similar DNA sequences in an inverted configuration at their terminals. Rather, each end contains a distinct ~150 base pairs (bp) long sequence with an absolutely conserved dinucleotide at the end of left terminal sequence (LTS), and a tetranucleotide at the end of right terminal sequence (RTS) which is preceded by a palindromic sequence that can form a hairpin structure (Fig. 1a). *Helitrons* integrate precisely between 5′-A and T-3′ nucleotides into host AT target sites, but do not generate target site duplications.

*Helitrons* encode a large (~1800 aa) multidomain transposase that does not contain an RNase H-like catalytic domain. Instead, an HUH endonuclease Rep domain is followed by a C-terminal helicase (Fig. 1a) related to the superfamily IB helicases that unwind DNA in the 5′–3′ direction. These two catalytic domains are preceded by an N-terminal DNA binding domain[21]. HUH nuclease domains (reviewed in ref. [22]) have one or two active site tyrosine residues that form a covalent 5′-phosphotyrosine linkage with DNA upon cleavage, while liberating the adjacent 3′-OH group. HUH nucleases are widely employed to carry out DNA transactions, such as rolling circle replication, viral replication initiation, conjugation, and DNA transposition, always acting on single-stranded DNA (ssDNA) and always with conserved strand polarity.

In prokaryotes, insertion sequence families, such as IS91 and the IS605/IS200 family, also rely on an HUH domain for the catalytic steps of transposition[23–25]. Experiments with IS91 have led to a "rolling circle" transposition model involving the excision

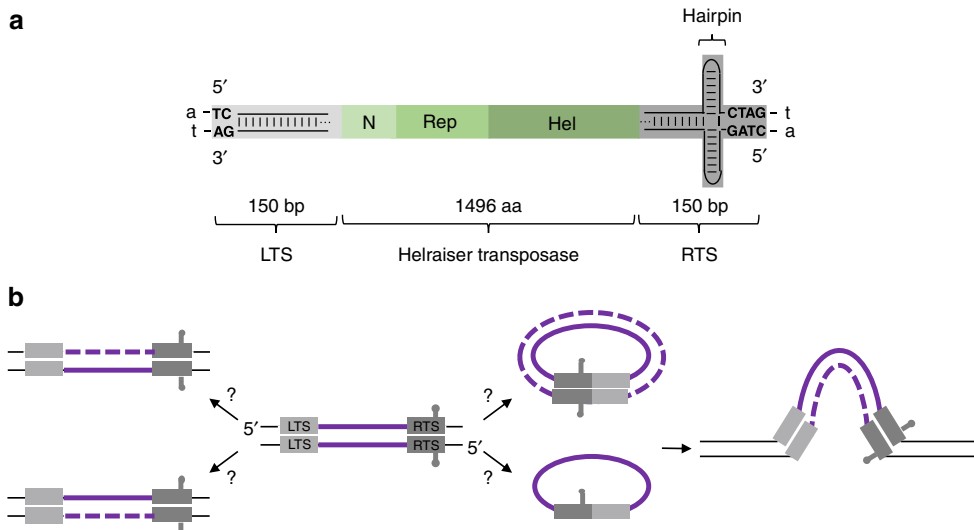

**Fig. 1** Features of *Helraiser* transposon and overview of transposition pathway. **a** Schematic representation of *Helraiser* transposon. The 150-bp left terminal sequence (LTS) is shown in light gray, and the 150-bp right terminal sequence (RTS) in dark gray, with the location of a potential hairpin indicated. N-terminal, Rep and helicase domains of the *Helraiser* transposase are shown as green rectangles. Conserved di- and tetranucleotide terminal sequence motifs are in uppercase. Flanking host A-T dinucleotide is in lowercase. **b** Schematic model and open questions of *Helitron* transposition. The transposase is proposed to excise one of the transposon strands from the double-stranded donor molecule (shown in the middle). Transposon excision is most likely followed by DNA synthesis that regenerates the transposon donor site (shown on the left). The excised transposon strand forms a single or double-stranded transposon circle that is possibly integrated into the target site in the host genome (shown on the right). LTS is shown in light gray and the RTS in dark gray. Solid purple lines: transposon donor strands; black lines: genomic sequence; and dashed lines: synthesized transposon strands

and reintegration of only one transposon strand, followed by DNA synthesis at the donor site[26]. This model has also been proposed for *Helitrons*[15], a simplified version of which is shown in Fig. 1b. However, IS*91* transposases do not encode a helicase, their terminal structures differ from those of *Helitrons*[23,24], and they integrate with site specificity[27]. Covalently closed transposon circles have been observed during IS*91* transposition[28], but their exact nature and whether they are intermediates that can act as substrates for integration remains elusive.

The best-studied prokaryotic HUH nuclease-encoding transposon is IS*608*[25,29]. Like other members of the IS*605*/IS*200*

family, it does not encode a helicase, it integrates with site specificity, and relies on DNA replication to make transposition substrates available in ssDNA form[30,31]. Like *Helitrons*, neither IS*91* nor IS*608* generate target site duplications, which probably reflects the inherently ssDNA nature of these transposition processes, since target site duplications are a consequence of transferring two transposon strands into opposite strands of target DNA at staggered positions which are subsequently repaired by a gap-filling mechanism.

Supporting the notion of rolling circle transposition by *Helitrons* are the results of data-mining that indicate head-to-tail

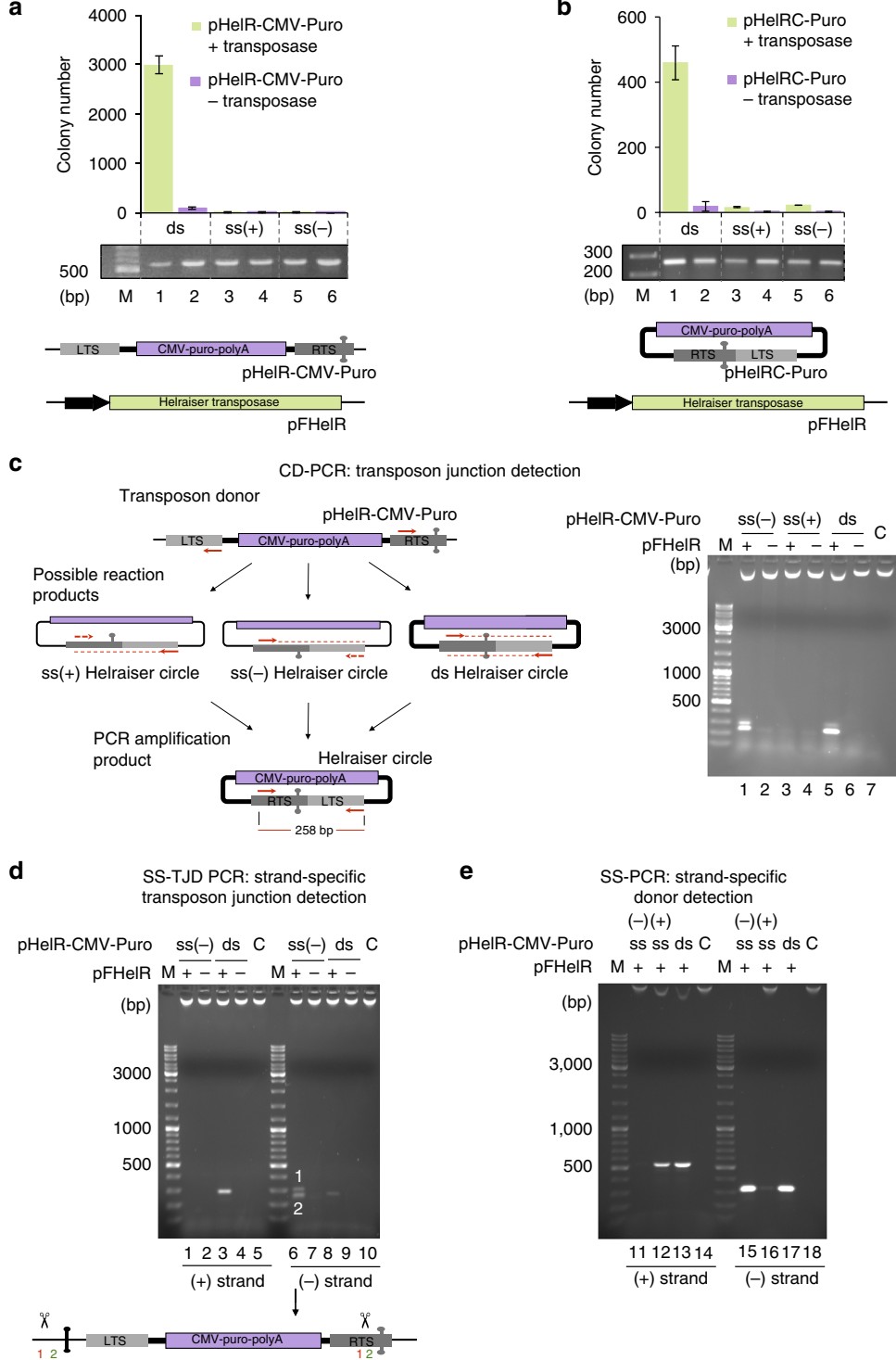

transposon junctions and tandem *Helitron* arrays in plant and bat genomes[17,18,32,33]. These features are reminiscent of the rolling circle replication of certain ssDNA viruses that is often accompanied by the formation of concatamers and multimeric DNA[34,35].

To experimentally investigate *Helitron* transposition, Grabundzija et al.[21] recently bioinformatically reconstructed an active *Helitron* transposon called *Helraiser* from ancient inactive copies of *Helibat1* transposons found in the little brown bat genome (Fig. 1a). This study demonstrated that *Helraiser* transposition generates covalently closed transposon circles containing a precise junction of the transposon ends in which the 5′-TC dinucleotide of the LTS is directly joined to the CTAG-3′ tetranucleotide of the RTS. Furthermore, double-stranded DNA (dsDNA) *Helraiser* circles propagated in *Escherichia coli* and transfected into HeLa cells were donors for integration, suggesting that transposon circles are transposition intermediates. Using active site point mutations of the HUH motif and the two tyrosines in the Rep domain (Y727 and Y731), as well as mutations in the Walker A and arginine finger motifs in the helicase domain, it was established that both the nuclease and helicase activities were needed for transposition in HeLa cells. In vitro, the HUH motif and Y731 were required for cleavage of the ssDNA oligonucleotides representing transposon ends. Results using transposon-end deletion mutants demonstrated that an intact LTS was crucial for transposition, whereas the RTS was not strictly required (although deletion of either RTS hairpin sequence or the entire RTS decreased the transposition efficiency). This study also provided insight into the mobilization of genes adjacent to the RTS, indicating that the hairpin structure serves as the transposition terminator and, as such, plays an important role in regulating gene capture. Collectively, these results provided a basis for experimental work on *Helitron* transposition, but many features of the transposition mechanism remained unclear.

As *Helitron* transposition is fundamentally different from cut-and-paste transposition, many questions arise, some of which are outlined in Fig. 1b. Can the donor site be a source of transposon DNA for multiple transposition events? Must the donor site be in dsDNA form or does ssDNA donor suffice? If transposition relies on ssDNA, which strand of the donor is transposed? Are *Helitron* circles generated in mammalian cells transposition intermediates? Are these present in ssDNA or dsDNA form?

Here, we demonstrate that the *Helraiser* donor site must be in dsDNA form and that ssDNA donors do not support transposition. Furthermore, for circular intermediates to serve as substrates for subsequent chromosomal integration, they must also be in the dsDNA form. In vitro, the *Helraiser* transposase forms precise transposon junctions but only from the top strand of the donors, and it also cleaves these top strand junctions precisely.

## Results

### *Helraiser* transposition from ssDNA and dsDNA donors.
Since all known HUH endonucleases cleave and join ssDNA substrates[22], we sought to establish if *Helraiser* transposition in mammalian cells requires ssDNA or dsDNA donor substrates. Furthermore, although it has been experimentally established that dsDNA *Helraiser* circles propagated in *E. coli* can be used as transposon donors in HeLa cells[21], it remained unclear whether the original transposon circles generated during transposition in HeLa cells were in single- or double-stranded form and whether those transposon circles were substrates for the *Helraiser* transposase. To address these questions, we used our previously described bicomponent transposon system[21] consisting of two plasmids: a donor plasmid encoding the transposon LTS and RTS and another expressing the *Helraiser* transposase. To quantify transposition events in a colony-forming assay, we tagged the *Helraiser* transposon with a cytomegalovirus (CMV) promoter-driven Puromycin (*Puro*) selection cassette in pHelR-CMV-Puro donor plasmids (Fig. 2a, bottom). To compare the ability of *Helraiser* to use ssDNA or dsDNA donors, we started from the dsDNA transposon donor, pHelR-CMV-Puro, and generated the two ssDNA donor versions by specifically nicking either the plus or the minus strand and then removing the nicked strand by exonuclease III digestion. We used the same procedure to generate ssDNA forms of the pHelRC-Puro transposon circle[21] which is, in essence, a circular transposon that contains a precise head-to-tail junction of the *Helraiser* ends (Fig. 2b, bottom).

We co-transfected dsDNA or ssDNA pHelR-CMV-Puro transposon donors to HEK293T cells either with the pFHelR *Helraiser* transposase helper plasmid or with a similarly sized control plasmid that did not express transposase. After 48 h, a portion of the transfected cells was subjected to *Puro* selection, while the rest were used to isolate the low-molecular weight (LMW) DNA fraction. This fraction was expected to contain both plasmids of the bicomponent assay, as well as any products that formed from the donor substrate.

The colony-forming assay (Fig. 2a, top) showed that co-transfection of the ssDNA pHelR-CMV-Puro donors and *Helraiser* helper plasmids did not yield more HEK293T *Puro*-resistant colonies than a negative control without transposase, while the dsDNA pHelR-CMV-Puro donor produced ~3000 *Puro*-resistant colonies per plate indicating robust transposition into genomic DNA. Similarly, transfection of HEK293T cells with dsDNA or ssDNA pHelRC-Puro transposon circles showed a stark contrast between the dsDNA and ssDNA forms (Fig. 2b,

**Fig. 2** *Helraiser* transposition from single- and double-stranded donors in human HEK cells. **a** Top: *Helraiser* transposition efficiency from double-stranded (ds) and single-stranded (ss) transposon donors, measured by *Puro*-resistant colony formation in HEK293T cells. All data are presented as a mean ± s.e.m., $n = 3$ biological replicates. M marker, ds transfected ds donors, ss(+) transfected plus-strand ss donors, ss(−) transfected minus-strand ss donors. Middle: PCR detection of transfected ds and ss transposon donors. All PCRs are performed with low-molecular weight (LMW) DNA isolated 48 h post-transfection. Bottom: schematic of the relevant portions of *Helraiser* donor (pHelR-CMV-Puro) and helper (pFHelR) plasmids. All plasmids are used in closed, circular form unless otherwise stated. Uncropped gel image is provided in Supplemental Fig. 1a. **b** Top: *Helraiser* transposition efficiency from ds, ss (+), and ss(−) transposon circles, as measured by *Puro*-resistant colony formation in HEK293 cells. ds transfected ds transposon circles, ss(+) transfected plus-strand ss transposon circles, ss(−) transfected minus-strand ss transposon circles. Middle: PCR detection of transfected ds and ss transposon circles. Bottom: schematic of the *Helraiser* circle (pHelRC-Puro) and helper (pFHelR) plasmid. Uncropped gel image is provided in Supplemental Fig. 1a. **c** *Helraiser* junction formation from ds and ss transposon donors. Left: outline of the CD-PCR method. Red arrows: forward (fwd) and reverse (rev) primer binding sites; dashed red line: first PCR cycle amplification products; dashed red arrows: primer binding sites on amplified products; full red line: final ds amplification product. Right: junction detection by CD-PCR. C no template control. Co-transfection of transposon donors with transposase helper plasmids is indicated as pFHelR "+." pFHelR "−" indicates co-transfection with control plasmids. **d** SS-TJD PCR of transposon circles. Lanes 1–5: PCR detects plus strand. Lanes 6–10: PCR detects minus strand. In lane 6, two sequenced PCR products containing *Helraiser* end junctions are indicated. Bottom: positions of the aberrant cleavage sites on the transposon donor molecule prior to end joining are indicated. **e** SS-PCR of transposon donors. Lanes 11–13, PCR detects plus strand; lanes 15–17, PCR detects minus strand

top); whereas the dsDNA circle was an effective substrate for transposition, yielding more than 400 *Puro*-resistant colonies per plate, ssDNA circles did not yield more than ~20 *Puro*-resistant colonies per plate, higher than their respective controls without transposase but lower than the dsDNA negative control.

To rule out the possibility that the observed differences in transposition efficiency between dsDNA and ssDNA donors and circles were due to transfection differences or the disappearance of ssDNA over time, we analyzed the isolated LMW DNA posttransfection with PCR. Using primers specific for the donor

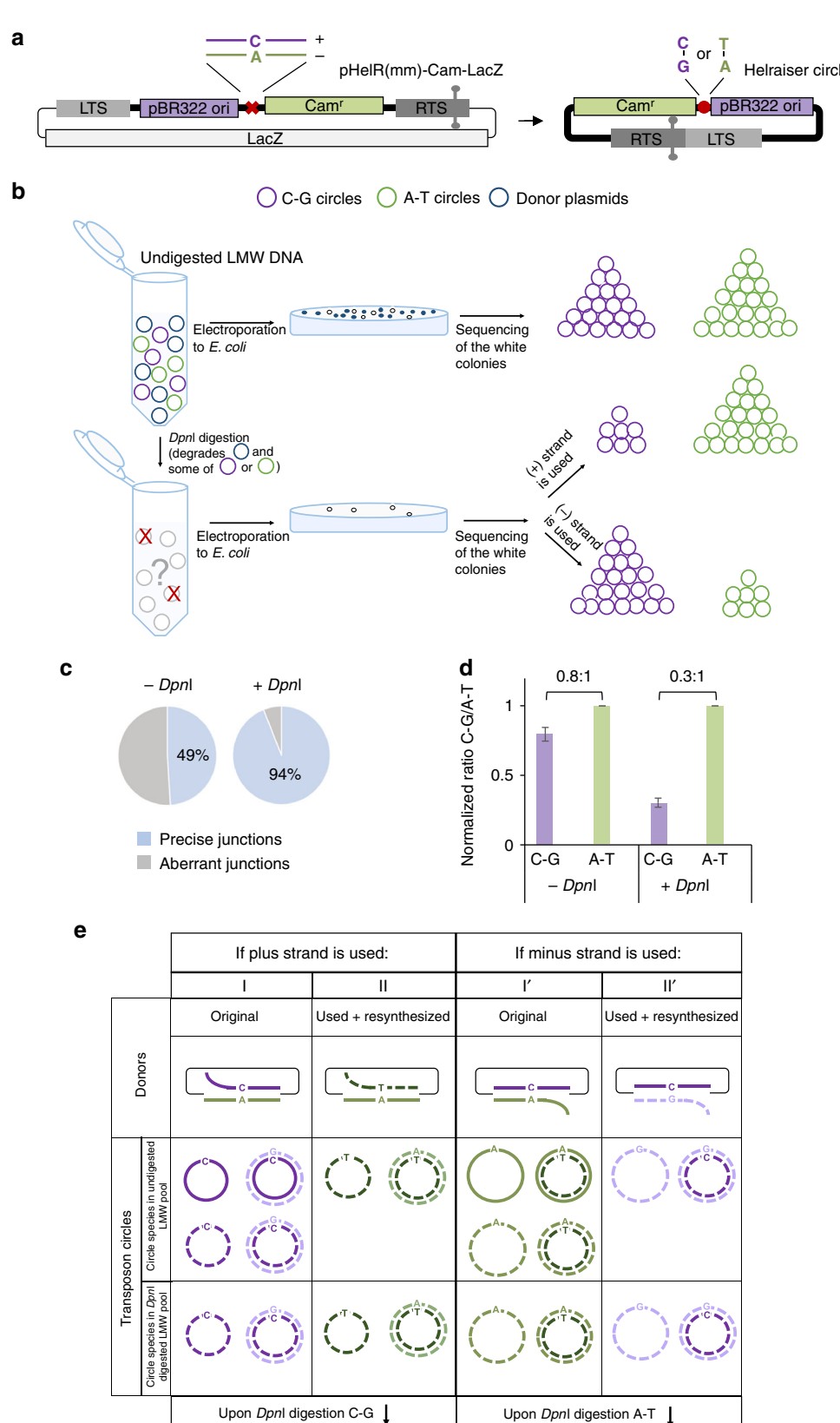

backbone + LTS or a primer pair specific to the LTS + RTS, all the various transposon forms assayed could be detected in cells 48 h posttransfection (Fig. 2a, b, lanes 1–6), indicating that the lack of *Puro*-resistant colonies after ssDNA transfection was not the result of either inefficient transfection or ssDNA degradation.

Given that dsDNA donors generate transposon circles[21], we asked whether ssDNA donors can also serve as substrates for the generation of *Helraiser* circles. As the circles contain a RTS-to-LTS junction, we used our transposon junction detection PCR (CD-PCR) method with the isolated LMW DNA as the PCR template. This detects the presence of RTS-to-LTS junctions indicative of *Helraiser* transposon-end joining but does not discriminate between ssDNA and dsDNA templates (Fig. 2c, left). As shown in Fig. 2c, lane 5, we observed junction formation in cells transfected with the dsDNA transposon donors, as expected. Surprisingly, we also detected CD-PCR products of a size consistent with circular transposon intermediates when the minus-strand ssDNA donors were transfected in the presence of transposase (Fig. 2c, lane 1) but not with plus-strand ssDNA donors (Fig. 2c, lane 3). This result suggested that the *Helraiser* transposase is able to catalyze junction formation using a minus-strand ssDNA donor.

As the results in Fig. 2c do not indicate which strand (or strands) are used to generate the circular intermediates, we developed a strand-specific transposon junction detection PCR (SS-TJD PCR) assay (Supplementary Fig. 1) to allow us to determine which transposon strand was used to form transposon-end junctions. When dsDNA transposon donors were transfected together with the *Helraiser* transposase plasmid, SS-TJD PCR detected transposon junctions of both the plus- and minus-strand polarity (Fig. 2d, lanes 3 and 8). In contrast, when minus-strand ssDNA donors were transfected with the transposase expressing helper plasmid, only minus-strand RTS-to-LTS junctions were detected (Fig. 2d, lanes 6 vs. 1). Thus, when dsDNA transposon donors are used, junctions of both polarity were formed, whereas minus-strand ssDNA donors yielded only minus-strand junctions. Although this result was informative and indicated that transposon circles of both strand polarities were made in HEK293T cells, it did not establish whether those circles were in dsDNA or ssDNA form.

Sequencing the two SS-TJD PCR products obtained with minus-strand ssDNA donors provided hints as to the reaction pathway; whereas the lower band (Fig. 2d, lane 6) contained some precise transposon-end junctions, both PCR bands were dominated by aberrant junctions. In these, DNA upstream from the LTS (i.e., within the plasmid backbone) had been joined to DNA within the RTS (Fig. 2d, bottom), indicating decreased fidelity of junction formation.

The possibility existed that ssDNA donor plasmids may have been converted to dsDNA form under our assay conditions. To test for this, we used a modification of the strand-specific PCR assay, SS-PCR, to determine the cellular fate of ssDNA donor plasmids. As a control for the assay, both the dsDNA pHel1-CMV-Puro donor strands were effectively detected when the

dsDNA donor was used to transfect (Fig. 2e, lanes 13 and 17), demonstrating that the donor persists in the transfected cells. In contrast, when the ssDNA donors were used, only the minus strand could be detected with the minus-strand donor (lanes 11 vs. 15) and, conversely, only the plus strand was detected with the plus-strand donor (lanes 12 vs. 16). Thus, within the time frame of the transfection experiment, we saw no detectable conversion of the ssDNA donors to dsDNA form. This is consistent with our earlier results (Fig. 2a) that ssDNA donors did not lead to transposon integration, as would have been expected had they been converted to the transposition-competent dsDNA form.

Collectively, these results suggest that *Helraiser* donor DNA must be in the dsDNA form to achieve transposition and for efficient transposon circle formation containing precise RTS-to-LTS junctions. The plus-strand ssDNA donors and transposon circles seem to be inactive, while the minus-strand donors produced predominantly aberrant junctions (Fig. 2c and d). Minus-strand ssDNA donors were not detectably converted to dsDNA, suggesting that misdirected reactions leading to aberrant junction formation occurred on ssDNA.

**Repair and reuse of *Helraiser* donor sites**. To establish which strand is transposed by *Helraiser*, we used a transposon donor containing a single base mismatch that allowed us to follow the fate of each transposon strand. We first constructed a "circle-rescue" transposon donor, pHelR(mm)-Cam-LacZ (Fig. 3a), in which the *Helraiser* LTS and RTS flank a Chloramphenicol (*Cam*) selection cassette and a bacterial replication origin (*pBR322 ori*) to allow the propagation in *E. coli* of *Helraiser* circles recovered from transfected cells. In order to use white/blue colony screening to identify *E. coli* clones containing *Helitron* circles, a *LacZ* marker was also placed in the donor plasmid backbone. To separately follow each transposon strand, we introduced a C-A nucleotide mismatch immediately downstream of the *pBR322 ori* with C in the plus strand (purple, Fig. 3a) and A in the minus strand (green). In this way, if only one strand is used for transposition, a dsDNA *Helraiser* circle generated from the plus strand from the original transposon donors and propagated in *E. coli* should contain a C-G bp at the mismatch site; whereas a circle generated from the minus strand should have an A-T bp at the same position, provided that the complementary strand of the circle is generated by replication, either in HEK293T cells or in *E. coli*.

The mismatched transposon donor was then transfected into HEK293T cells. The mismatch introduced in the *Helraiser* donor was expected to be maintained after transfection since the pHelR(mm)-Cam-LacZ plasmid should not replicate or be repaired in the mismatch repair deficient HEK293T host[36]. We confirmed this by sequencing the transposon donors prior to and after transfection, and also by fluorescence-activated cell sorting (FACS) analysis.

To determine which of the two bases of the mismatch was present in the transposition circle intermediate, LMW DNA was recovered 48 h posttransfection, electroporated into *E. coli*, and *Helraiser* circles were isolated from white (*LacZ*−) *Cam* resistant

**Fig. 3** *Helraiser* circle replication and donor site repair in HEK293T cells. **a** Schematic of the *Helraiser* heteroduplex *LacZ* donor plasmid (pHelR(mm)-Cam-LacZ) and resulting *Helraiser* circle. The red cross indicates the mismatch position within the transposon sequence, and the red circle marks the position of the mismatch used in the analysis of the *Helraiser* circles. **b** Experimental design of transposon circle replication assay using heteroduplex pHelR(mm)-Cam-LacZ donor plasmid. As shown, *Dpn*I digestion of LMW DNA reaction products can be used to distinguish between transposition of the (+) strand and the (−) strand. **c** Proportion of the transposon circles containing precise LTS-to-RTS junctions before and after *Dpn*I digestion of electroporated LMW DNA. The data are presented as n = 3 biological replicates. **d** Results of the transposon circle replication assay with pHelR(mm)-Cam-LacZ plasmid. The data are presented as a mean ± s.e.m., n = 3 biological replicates. **e** Schematic representation of possible outcomes of the transposon circle replication assay with heteroduplex pHelR(mm)-Cam-LacZ donors. Purple line: (+) strand of transposon donor; green line: (−) strand of transposon donor; solid line: methylated DNA; dashed line: unmethylated DNA; thin black line: plasmid backbone

(*Cam*+) *E. coli* colonies. These were then analyzed with respect to the formation of precise RTS-to-LTS junctions and the bp (C-G or A-T) at the position corresponding to the donor mismatch site (as illustrated in Fig. 3b, top row). Sequencing of *Helraiser* circles revealed that 49% contained precise RTS-to-LTS junctions (shown in blue, Fig. 3c). Surprisingly, both C-G and A-T bp were present at the mismatch position: isolated circles consisted 55% with A-T and 45% with C-G at the mismatch position (Fig. 3d) (We cannot rule out the possibility that some of the transposition reaction products might be unstable in HEK293T cells, potentially introducing biases into our analysis.).

If all the recovered *Helraiser* circles had originated from the original donor molecules, they would have exclusively either C-G (if the top strand is transposed) or A-T (if the bottom strand is transposed) at the mismatch position as shown in Fig. 3e I and I′, but this is not what we observed. Furthermore, we have not detected empty donor sites indicative of double-stranded *Helraiser* excision. Thus, the most parsimonious explanation of our results is that resynthesis and reuse of the transposed strand has "scrambled" the C-A mismatch read-out within the resulting *Helraiser* circles as shown in Fig. 3e.

**Role of DNA replication in *Helraiser* circle formation**. To determine if DNA replication plays a role in the formation of *Helraiser* transposition intermediates, we employed a *Dpn*I-based assay[37–39]. This assay takes advantage of the *dam* methylation occurring in *E. coli* but not in eukaryotic cells. As all plasmid DNA used in transfections was propagated in *E. coli*, it contained *dam*-methylated GATC sequences. Restriction endonucleases *Dpn*I and *Mbo*I are either enabled or inhibited, respectively, by the hemi or full methylation of their recognition sequence (GATC); therefore, they can be used to identify and distinguish *dam*-methylated from unmethylated DNA[40], and hence *E. coli*-synthesized DNA from DNA synthesized in eukaryotic cells.

We combined a *Dpn*I assay and heteroduplex transposon donors to select for any *Helraiser* circles containing DNA synthesized in eukaryotic cells (Fig. 3b, bottom row). The experimental rationale was that any *Helraiser* circles formed during the transposition reaction that survive *Dpn*I digestion will contain only unmethylated DNA; hence, both strands must have been synthesized in HEK293T cells. As a test of our approach, we first used synthetic oligonucleotides to establish that *Dpn*I treatment indeed digested both *dam* hemimethylated and methylated dsDNA (Supplementary Fig. 2a), and also ssDNA and dsDNA forms of *Helraiser* circles. We then used the LMW DNA isolated from the same transfection described in the previous section but included an overnight digestion with *Dpn*I prior to electroporation as illustrated in Fig. 3b. In this way, only unmethylated *Helraiser* circles would give rise to white *Cam*+ *E. coli* colonies (To further confirm the lack of methylation, the products of the *Dpn*I reactions were digested with *Mbo*I, resulting in the complete loss of white colonies as expected (Supplementary Fig. 2b).).

According to the possible circle formation pathways those transposon circles that contain an original *dam*-methylated transposon donor strand would not survive *Dpn*I digestion. All those arising from repaired donors are comprised of DNA synthesized in HEK293T cells, and hence are unmethylated and would be unaffected by *Dpn*I digestion. Sequencing white *Cam*+ bacterial colonies obtained after electroporation of the *Dpn*I digestion product showed, consistent with what we observed without digestion, that both C-G and A-T bp were present at the mismatch position in *Helraiser* circles. However, *Dpn*I digestion markedly changed the ratio of C-G to A-T bp at the mismatch position from 0.8:1 to 0.3:1, respectively (Fig. 3d). The direction of the change in the ratio at the mismatch position of C-G to A-T bp after *Dpn*I digestion is a strong indicator of which strand was used to generate transposon circles. This is because if the plus strand of the donor molecule was used for transposon circle formation (Fig. 3e, group I and II), *Dpn*I digestion would remove those circles that had the original, methylated strand with the C base and, regardless of the distribution of the products, the proportion of circles containing C-G will decrease. Conversely, if the minus strand was used for transposition, *Dpn*I digestion would remove those circles that contain A-T and the proportion of circles containing C-G will increase (Fig. 3e, I′ and II′). The reduction that we observed of the relative proportion of circles with a C-G bp at the mismatch position after *Dpn*I digestion, therefore, suggests that the plus strand of the donor that was used for transposon circle formation. This result is in agreement with our previous observations[21] in vitro that the *Helraiser* transposase catalyzed precise cleavage of the ssDNA substrates representing both transposon ends only in the case of plus-strand oligonucleotides.

The isolation of any *Dpn*I-resistant *Helraiser* circles containing C-G at the mismatch position is indicative of strand synthesis within HEK293T cells in the context of the transposon circles. One way this could have occurred is if the transposon circles created through initial excision from the donor site were used as transposon donors. In this possible pathway, the dsDNA circles would be cleaved at the plus-strand junction by the transposase, initiating strand synthesis and ultimately resulting in the formation of completely unmethylated circles.

Another important observation provided by these experiments was the proportion of aberrant junctions before and after *Dpn*I digestion (Fig. 3c). Without *Dpn*I digestion, ~50% of the circles contained aberrant junctions, but in the screened *Dpn*I-digested colonies (i.e., those consisting of unmethylated DNA), 94% of the sequenced *Helraiser* circles contained precise LTS-to-RTS junctions. One possible interpretation of this result is that dsDNA circles containing a precise RTS-to-LTS junction are preferred substrates for subsequent *Helraiser* circle replication.

**Helraiser integration from mismatch-containing donors**. To investigate *Helraiser* genomic integration, we made another set of mismatch-containing transposon donors with a CMV-GFP-IRES-Puro cassette between the *Helraiser* LTS and RTS (Fig. 4a). In this arrangement, *GFP* allowed the visualization of transposon integration events in HEK293T cells, while the *Puro* cassette mediated selection and quantification of colonies harboring genomic *Helraiser* insertions. We also introduced a single base mismatch tag into the ATG start codon of the *GFP* gene in the pHelR-GFP-Puro plasmid, generating a heteroduplex donor vector (pHelR(mm)-GFP(−)-Puro (Fig. 4a, left)) in which the *GFP* start codon was exchanged from ATG to ACG only in the coding plus strand. In this way, the mismatch (and hence its consequence in terms of a resulting *GFP* signal) provides information on which transposon strand becomes integrated. As shown schematically in Fig. 4a, integration of the *Helraiser* plus strand of pHelR(mm)-GFP(−)-Puro into the genome of HEK293T cells would therefore result in *Puro*-resistant (*Puro*+) colonies, which would not express GFP (*Gfp*−) due to the lack of a start codon. In contrast, integration of the minus-transposon strand would yield *Puro*+/*Gfp*+ colonies.

As a positive control, we used the pHelR-GFP-Puro donor plasmid with an unmodified *GFP* coding sequence (Fig. 4b, first column). As a negative control, we created the pHelR-MutGFP-Puro donor vector by exchanging the *GFP* start codon from ATG to ACG (mutating both strands) that did not express *GFP* (Fig. 4b, column 4). As a further permutation, we constructed a

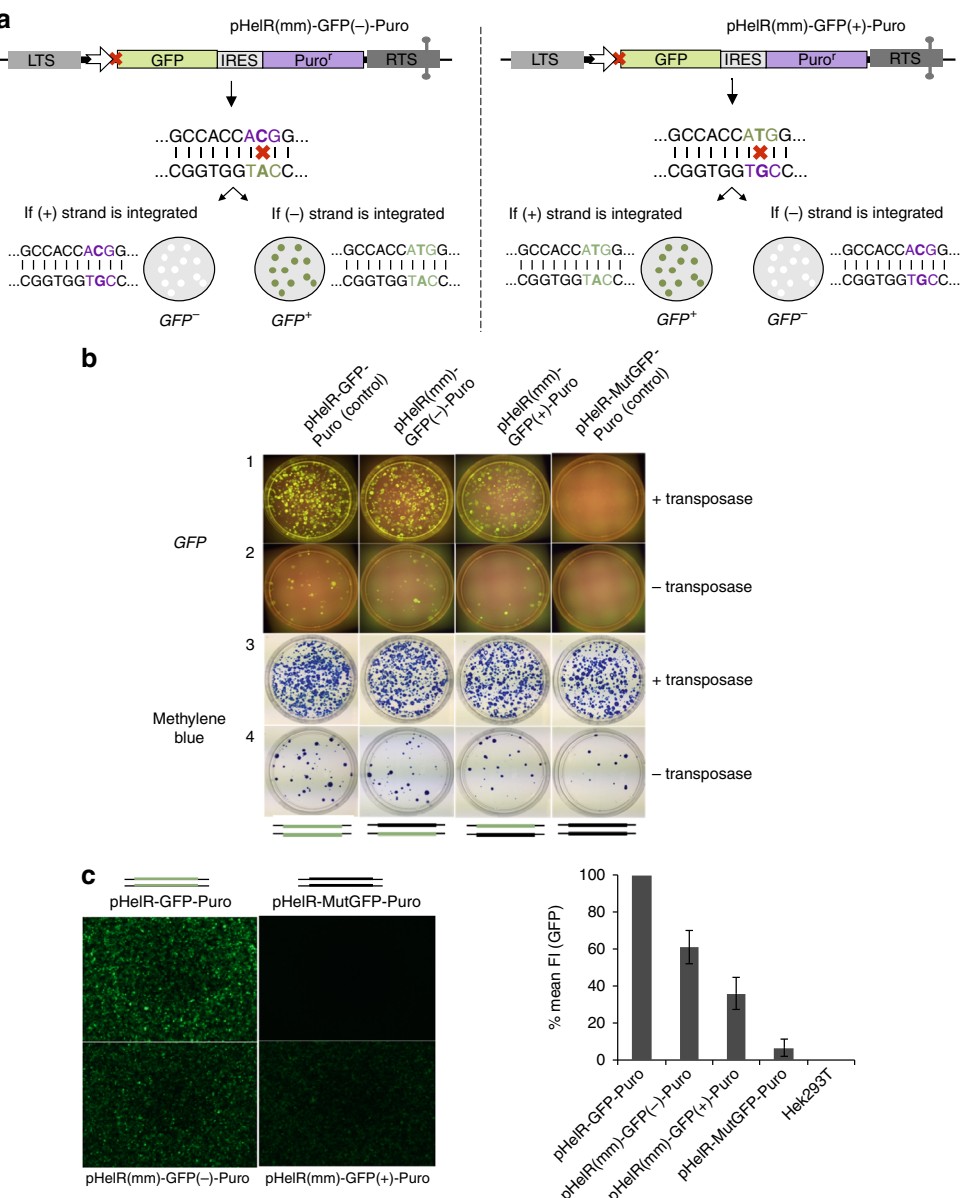

**Fig. 4** *Helraiser* transposition from heteroduplex *GFP-Puro* reporter plasmids in HEK293T cells. **a** Schematic of relevant portions of *Helraiser* heteroduplex *GFP-Puro* donor plasmids, pHelR(mm)-GFP(−)-Puro (left) and pHelR(mm)-GFP(+)-Puro (right). Shown are possible outcomes of *Helraiser* transposon integration for each of the two heteroduplex donors. Mismatch positions on transposon donors are indicated by red x with sequences shown below. Mutated sequence of the *GFP* start codon is in purple; intact *GFP* start codon in green. Schematic representations of tissue culture plates with *GFP* negative (*GFP*−) and *GFP* positive (*GFP*+) *Puro*-resistant colonies are as shown. **b** Colony-forming assay with the two heteroduplex and the two control transposon donors. Tissue culture plates containing *Puro*-resistant colonies 22 days post-transfection were first imaged under blue light (*GFP*) followed by methylene blue staining. **c** FACS analysis of the *GFP* fluorescence intensity (FI) in *Puro*-resistant Hek293T cells 22 days post-transfection. Left: fluorescence microscopy images show the *Puro*-resistant cell suspensions used for FACS analysis. The data are presented as a mean ± s.e.m., $n = 3$ biological replicates. Schematic of transposon donors indicate the transposon strand with mutated (thick black line) and the intact *GFP* start codon (thick green line); thin black lines: plasmid backbone

heteroduplex donor with inverted *GFP* expression properties: in pHelR(mm)-GFP(+)-Puro (Fig. 4a, right), the ATG *GFP* start codon was present in the coding plus strand of the heteroduplex transposon donor, while the noncoding minus strand had TGC. In this case, integration of the transposon plus strand was expected to lead to the generation of $Puro^+/Gfp^+$ colonies, while integration of the minus strand would lead to the $Puro^+/Gfp^-$ colonies that do not express *GFP*. However, if transposon donor site resynthesis and reuse occurred, we would expect to see *GFP* expression with both of the two heteroduplex donors. For

example, if the plus strand of the *pHelR(mm)-GFP(−)-Puro* heteroduplex donor was used, among the multiple integrations expected within the cell genome[21], insertions originating from the repaired donors would express *GFP*, whereas only those containing original transposon strand would not encode expression of the *GFP* gene (and similarly—but for the opposite strands—for the *pHelR(mm)-GFP(+)-Puro*). Alternatively, if donor site resynthesis and reuse were absent, all of the cells transfected with the individual heteroduplexes would either be $Puro^+/Gfp^+$ or $Puro^+/Gfp^-$.

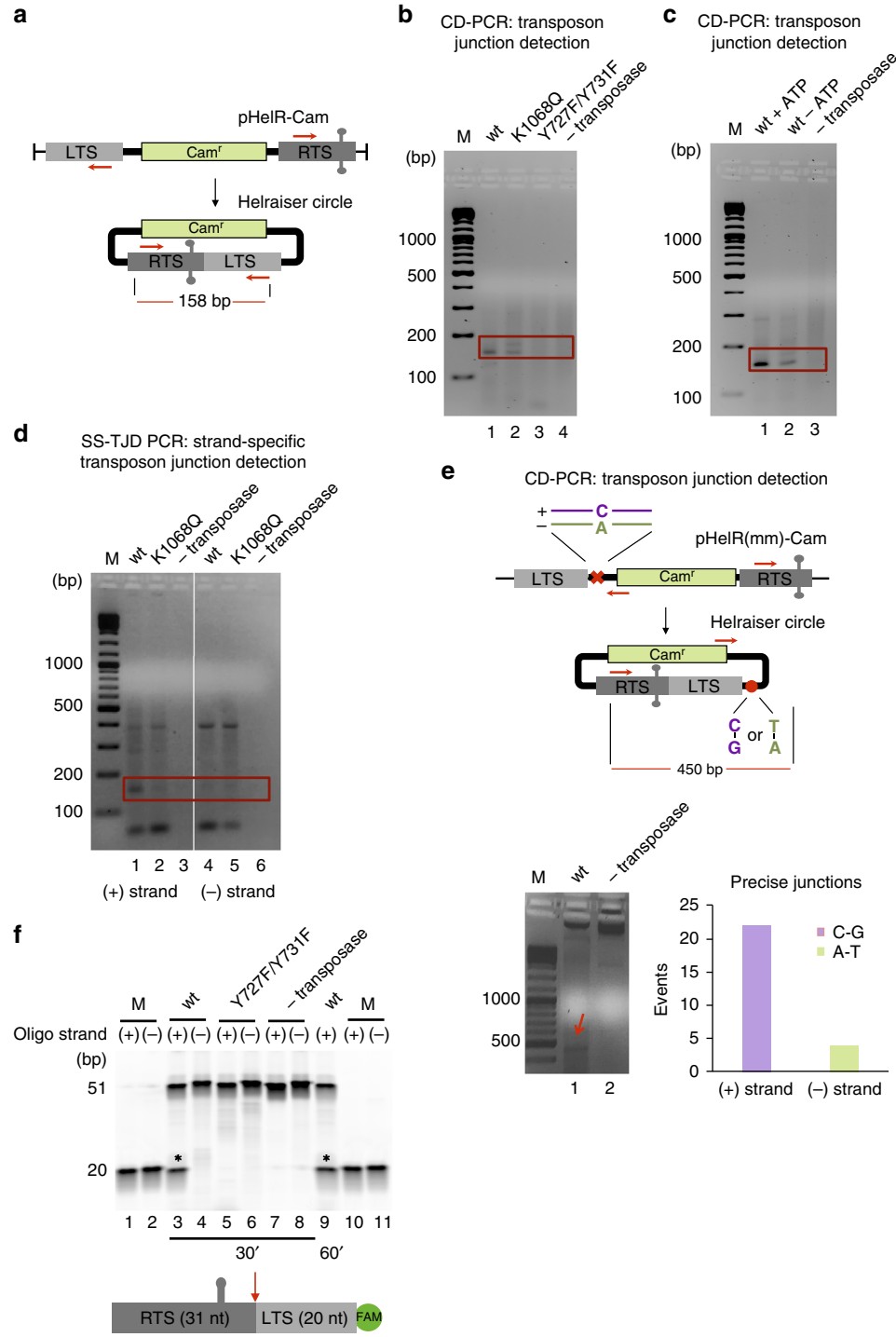

**Fig. 5** In vitro transposon-end junction formation and cleavage with purified *Helraiser* transposase. **a** *Helraiser* donor plasmid (pHelR-Cam), used in linear form, and resulting *Helraiser* circle. Red arrows represent fwd and rev nested primer binding sites; red line: expected size of the PCR product. **b** PCR detection of *Helraiser* transposon-end junctions. M marker, Red box marks the position of the expected PCR product. **c** PCR detection of *Helraiser* transposon-end junctions ±ATP. **d** Strand-specific PCR detection of transposon-end junctions generated in vitro by the *Helraiser* transposase. Lanes 1–3: detection of plus strand. Lanes 4–6: detection of minus strand. **e** Top: schematic of *Helraiser* heteroduplex donor plasmid, pHelR(mm)-Cam and resulting *Helraiser* circle. Red x mismatch position within transposon sequence; red circle position of the mismatch on the donor molecule used in the analysis of the *Helraiser* circles. Bottom left: PCR detection of the transposon-end junctions generated from the heteroduplex donor. Red arrow indicates PCR product of the expected size spanning the mismatch position. Bottom right: DNA base composition at the mismatch position in the obtained PCR product. Most of the detected junctions arose from the plus strand; four arose from the minus strand. **f** Top: cleavage of ss 51-mer DNA oligonucleotides representing plus and minus strand of transposon-end junction. Marker (M): 20 bp oligonucleotide corresponding to the expected cleavage product for each strand. Bottom: schematic representation of the ss oligonucleotide used in cleavage reactions. Red arrow indicates the cleavage site. Oligonucleotide sequences are listed in Supplementary table 1

Each of the heteroduplex transposon donors (pHelR(mm)-GFP(−)-Puro and pHelR(mm)-GFP(+)-Puro), as well as the two *GFP* expression control plasmids were transfected into HEK293T cells. As shown in Fig. 4b, there is robust *Helraiser* transposition when the transposase is present, as indicated by the *Puro*+ colonies (row3 vs. row4). Interestingly, regardless which of the two heteroduplex donors were transfected, fluorescent microscopy analysis of the *Puro*+ colonies 22 days post-transfection revealed that *Puro*+/*Gfp*+ colonies were generated (Fig. 4c, left). This is consistent with our previous observations made with *Helraiser* circles that the donor sites are resynthesized and reused.

FACS analysis of the *Puro*+ colonies (Fig. 4c, right) indicated that *GFP* expression was stronger when the pHelR(mm)-GFP(−)-Puro heteroduplex donor was co-transfected with the transposase helper plasmids than that observed with the pHelR(mm)-GFP(+)-Puro donor. This suggested that integration originated more frequently from the resynthesized than the original donor sites.

**In vitro formation of *Helraiser* circles**. We have previously demonstrated that purified *Helraiser* transposase cleaved ssDNA oligonucleotides representing the *Helraiser* LTS and RTS in an in vitro system in the presence of divalent metal ion and ATP cofactors[21]. We took advantage of our CD-PCR and SS-PCR assays to ask whether purified *Helraiser* cleaves dsDNA donors and form transposon junctions without cellular components.

We performed an in vitro transposon junction formation assay in the presence of ATP and metal ions ($Mg^{2+}$ and $Mn^{2+}$) using a linearized version of the pHelR-Cam transposon as the transposon donor (Fig. 5a). The products were detected with nested CD-PCR and visualized on an agarose gel (Fig. 5b). Sequencing of the PCR products showed that the wild-type (wt) *Helraiser* transposase (lane 1), but not the Y727F/Y731F mutant (lane 3), generated precise RTS-to-LTS junctions in vitro. A helicase Walker A motif mutant, K1068Q, formed some transposon RTS-to-LTS junctions (lane 2) although with much lower efficiency than the wt *Helraiser* transposase. Consistent with this, in the absence of ATP, junction formation by the wt transposase was significantly compromised but was not abolished (Fig. 5c). These results indicate that, in principle, no cellular proteins were required for precise junction formation in vitro, and while the nuclease activity of the transposase was required, the helicase activity was not absolutely required although it was stimulatory.

To determine which of the two transposon strands was used for RTS-to-LTS transposon junction formation in vitro, we used SS-CD PCR, as previously described for cell-based experiments (Supplementary Fig. 1). We obtained a PCR band of a size corresponding to *Helraiser* RTS-to-LTS junctions only when the primer specific to the plus, but not to the minus, strand was used in the first SS-CD PCR round (Fig. 5d, lanes 1 vs. 4). Sequencing confirmed that precise RTS-to-LTS junctions were formed. Formation of precise plus-strand junctions was consistent with our results from HEK293T cells.

We also performed the circle formation assay in vitro using the heteroduplex pHelR(mm)-Cam donors, and sequenced the products of the CD-PCR reactions (Fig. 5e, lane 1). Analysis of the precise RTS-to-LTS junctions showed that most of the events (22 of 26) contained C at the mismatch position, demonstrating that the vast majority of junctions were made using the plus strand of the transposon donors. We also recovered several imprecise junctions, four of which belonged to the plus and one to the minus strand. The four minus-strand products might have been the consequence of transposase cleaving and joining the exposed minus strand.

Finally, we carried out an in vitro cleavage reaction on short single-stranded oligonucleotides to compare the ability of the *Helraiser* transposase to cleave either the plus or minus strand of a precise RTS-to-LTS junction. As shown in Fig. 5f, only the plus strand is cleaved (lanes 3, 9), and no cleavage is observed of either strand when the active site double mutant Y727F/Y731F is used (lanes 5,6). The specific top strand junction cleavage reaction reaches a 50–50% ratio between substrate and product. This is the expected behavior of an HUH nuclease as one of the products stays covalently bound to the transposase through a 5′ phosphotyrosine linkage that can be resolved and the substrate state re-established by the free 3′OH end of the other product[25].

## Discussion

*Helitrons* are a unique group of DNA transposons that have profoundly impacted eukaryotic genomes and that transpose using a mechanism that differs from all other characterized eukaryotic DNA transposons. One key feature is the presence of covalently closed circular molecules containing precise RTS-to-LTS junctions. While such circular intermediates occur in a number of different prokaryotic transposition processes[28,41,42], they have not been seen before in any other eukaryotic system. Here we have addressed a number of fundamental mechanistic questions using *Helraiser*, an active *Helitron* transposon reconstituted from the genome of the bat *Myotis lucifugus*[21].

In order to examine the mechanism of *Helitron* transposition we developed novel methodologies to study eukaryotic replicative transposition. Combining strand-specific reporter assays, such as heteroduplex donors, with various prokaryotic and eukaryotic reporters (e.g., *LacZ*, *GFP*) allowed us to differentiate between the two transposon strands and to follow their fate during the transposition process. In addition, assays we adapted from viral systems, such as a *Dpn*I replication assay and strand-specific PCR, allowed us to address the questions involving DNA synthesis, such as transposon donor repair and the DNA form of transposon circles.

Despite the overwhelming body of evidence that HUH nuclease-containing enzymes perform cleavage and strand transfer exclusively on ssDNA substrates, we show here that for integration to occur in HEK293T cells, the transposon donor site must be in dsDNA form and that ssDNA donors will not suffice (Fig. 2a). We have previously shown that *Helraiser* circles are formed in HeLa cells from dsDNA donors and that transfected dsDNA *Helraiser* circles propagated in *E. coli* served as transposon donors for integration[21]. However, it was not clear whether the transposon circles generated during transposition in HeLa cells were in ssDNA or dsDNA form, nor if those transposon circles were actively involved in transposition process. Here we extend our previous findings to show that circular transposons must also be in the dsDNA form to act as intermediates for chromosomal integration (Fig. 2b). Furthermore, plus-strand ssDNA donors do not yield detectable junctions, whereas minus-strand ssDNA donors can lead to junction formation although most of them were aberrant (Fig. 2c). These results suggest a critical role for dsDNA in *Helitron* transposition, perhaps because the *Helraiser* transposase can initiate transposition only in the context of dsDNA or at a dsDNA/ssDNA junction. Interestingly, circular ssDNA viruses (gemini-, nano, circo-) that use rolling circle replication show a similar requirement as their ssDNA genomes first have to be converted into the dsDNA form for replication to start (reviewed in refs. [43–45]). In these systems, the viral initiator recognizes the viral origin only in the dsDNA form, followed by rolling circle replication to generate multiple viral genomes (reviewed in refs. [43–45]). Perhaps, it is not a coincidence

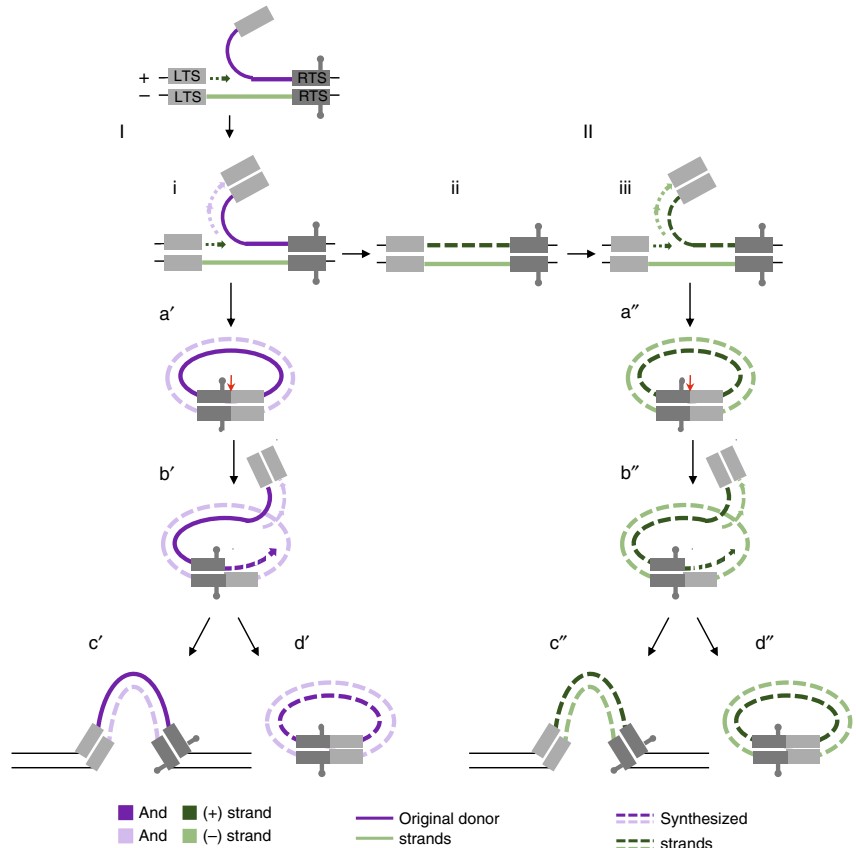

**Fig. 6** Proposed pathways of transposon circle formation together with replication and integration during *Helraiser* transposition. Transposon circle formation using the original donor site is shown on the left (I). Following the nicking of transposon plus strand at the LTS, a replication fork is established at the nick site (i). Leading strand synthesis reconstitutes the transposon donor site (ii), while lagging strand synthesis takes place on the displaced transposon strand. This results in a formation of dsDNA transposon circle (a′) after the second cleavage reaction takes place at the RTS of the displaced transposon strand. Generated dsDNA transposon circle potentially serves as a transposon donor (b′). Following the transposon plus-strand cleavage at the LTS-to-RTS junction site on a dsDNA transposon circle, a replication fork is established at the nick site (b′). Leading strand synthesis regenerates the transposon donor site on the transposon circle, while the lagging strand synthesis results in a dsDNA transposon copy that can be integrated in the host genome (c′) or form a new dsDNA transposon circle (d′). After resynthesis of the original transposon donor site (shown on the right (II)), transposon circle formation and replication include the same steps illustrated for the pathway (I). Solid line: original transposon donor strand; dashed line: synthesized transposon strand. Transposon plus strand is shown in dark purple or in dark green; transposon minus strand is shown in light purple or in light green. Red arrow marks the position of the transposon plus-strand cleavage at the LTS-to-RTS junction site

that these viral replication initiators also consist of an HUH nuclease Rep domain followed by a helicase domain.

The rolling circle model of transposition proposes that only one strand of the transposon is excised and reintegrated, yet this had not been previously experimentally demonstrated. Our data here suggest that it is the plus strand of *Helraiser* that is active. We conclude this from the data obtained in HEK293T cells with heteroduplex substrates, and also from the in vitro experiments with purified *Helraiser* transposase, which indicated that only plus-strand junctions could be detected when the transposase was incubated with a linear dsDNA plasmid donor, and that ~85% of the precise product junctions arose from the plus strand when the dsDNA donor contained a mismatch (Fig. 5e). Furthermore, recapitulating junction cleavage in vitro on ssDNA oligonucleotide substrates revealed that only the plus strand can be cleaved and that it is cleaved precisely at the junction point; this is a key step in the rolling circle model for the generation of a cleaved transposon end from circular intermediates for subsequent integration. The apparent discordance between efficient ssDNA cleavage in vitro (Fig. 5f) and the apparent need in cells for dsDNA circular transposon intermediates (Fig. 2b) may hint at the role of the *Helraiser* helicase domain, which may be needed to

locally provide limited regions of ssDNA that can serve as the substrate for the endonucleotic reaction by the HUH domain to generate the 3′OH replication start point.

A key finding in our experimental system is the observed reuse of the original donor site, which in principle provides a constant source of donor DNA supporting multiple rounds of transposition. Our *Dpn*I data also suggest an enrichment of the precise transposon junctions when looking at only newly synthesized DNA. This result is consistent with a model in which the precisely joined circles themselves act as an additional sources of new circular intermediates, or as transposon donors for integration. These data confirm that *Helraiser* circles generated during the transposition are actively involved in this process and are bona fide transposition intermediates, a result that had been previously only inferred[21]. Moreover, our findings reveal the potential of *Helraiser* circles to act as transposon donors for further rounds of circle formation in the presence of the transposase. In fact, a parallel can be drawn with certain ssDNA viruses, whereby repeated transposon donor reuse and resynthesis, as well as the multiple rounds of DNA synthesis on transposon circles could be equated to the replication process. In addition, the possibility that the episomal forms of *Helraiser* transposon could replicate and

persist in the cells in the presence of transposase points to the potential of this transposon system.

Our data demonstrated that only the plus strand of the transposon donor molecule is transposed. This would lead to formation of ssDNA transposon circles of the plus-strand polarity. However, neither transfection of the plus strand ssDNA donors nor ssDNA *Helraiser* circles resulted in transposon integration in HEK293T cells (Fig. 2a, b). This indicated a previously unsuspected feature of the mechanism that transposon donors and circles have to be in dsDNA form to serve as integration substrates, yet we did not detect any ssDNA to dsDNA conversion when ssDNA donors or circles were transfected, another curious facet of *Helitron* transposition (Fig. 2d, e). In contrast, our data demonstrated that at least a portion of *Helraiser* circles generated in HEK293T cells were in dsDNA form (Fig. 3d). This discrepancy can be explained if a replication fork is formed at the donor site after 5′-end cleavage of the top strand as illustrated in Fig. 6. In this manner, leading strand synthesis would reconstitute the donor site (Fig. 6i, ii), while lagging strand synthesis would take place on the peeled-off transposon strand (Fig. 6i), as demonstrated in the rolling circle replication of certain dsDNA viruses and phages[46–48]. Since the peeled-off strand is used to generate transposon circles, through this process they would become double stranded without ever forming a free ssDNA circle (Fig. 6a′).

Another possible model to explain our results draws from the replication of Circoviruses[34]. During the replication process of these circular ssDNA viruses, a short minus-strand primer is made on the replicated plus-strand viral genome and thus primed viral genomes are packaged into virions. Upon virus infection, the minus-strand primer is used for minus-strand synthesis that leads to formation of dsDNA replicative intermediates[34]. If *Helitrons* use a similar mechanism for minus-strand synthesis and conversion of the plus-strand ssDNA into dsDNA transposon circles, the unprimed ssDNA circles transfected in our experiments would not be converted to the dsDNA form, and would thus remain transpositionally inactive.

Our current in silico reconstructed transposase, while active both in vivo and in vitro, has low activity in vitro and is likely to represent a suboptimized version. It is also possible, as it is the case with ssDNA and dsDNA viruses[43,49–51], that in order to exhibit optimal activity the *Helraiser* transposase needs to interact with components of the cellular replication or repair machinery.

Our results provide strong support of a transposition model in which steps of DNA cleavage and strand transfer occur only and specifically on a single transposon strand, but depend on a framework provided by dsDNA, tightly linking *Helitron* transposition to DNA synthesis and generation of dsDNA. This, and the use of circular dsDNA transposition intermediates, suggests that *Helitron* transposition shares intriguing similarities with the replication of circular ssDNA viruses.

## Methods

**Constructs**. Detailed cloning procedures of transposon donor plasmids are provided in Supplementary Note 1.

**Heteroduplex plasmid generation**. Design of the mismatch region and the heteroduplex generation procedure (Supplementary Fig. 3) followed the previously described method[52,53] with modifications. Briefly, in all heteroduplex donors used in this study, a portion of the plus strand, flanked by two *Nt.BbvCI* enzyme (NEB) nicking sites, was replaced with the phosphorylated mismatched oligonucleotide (Supplementary Table 1 lists the oligonucleotides used) that was ligated into the replaced region. Detailed description of the mismatch region and heteroduplex generation procedure can be found in Supplementary Note 2.

**Single-stranded plasmid generation**. To generate ss transposon donor plasmids, we applied previously described methods[54–56] with some modifications. Briefly,

one strand of the supercoiled plasmid DNA (~40 µg DNA) was incubated with a nicking enzyme overnight. This was followed by Exonuclease III (NEB) digestion to degrade the nicked strand and generate ssDNA. Detailed description of the ssDNA generation procedure can be found in Supplementary Note 3.

**PCRs**. LMW DNA was isolated from transfected HEK293T cells as previously described[21]. Unless stated otherwise, 200 ng of the isolated LMW DNA and 200 nm primers (each) were used per 50 µl PCR reaction. Detailed PCR procedures are described in Supplementary Note 4. PCR primer sequences are listed in Supplementary Table 1.

**Cells and transfection**. HEK293T cells (gift of M. Gellert lab) ($1 \times 10^6$) were seeded onto six-well plates 1 day before transfection. For transfection of ssDNA and dsDNA forms of pHelRC-Puro transposon circles (Fig. 2b), HEK293 cells were used. All the transfections were performed with Lipofectamine 3000 according to manufacturer's protocol. Typically, 500 ng of dsDNA or ssDNA transposon donors and 500 ng of helper plasmids were transfected to the cells. As a negative control, in all transfection experiments, pFHelR helper plasmids were replaced with pLexNHH plasmids of similar size. In the experiments with *Gfp-Puro* donors, 150 ng of transposon donor plasmids were co-transfected with 150 ng of helper or control plasmids. Forty-eight hours post-transfection, fraction of the cells was replated onto 100 mm dishes and selected for transposon integration with Puro (2 µg/ml). Alternatively, LMW DNA was isolated from the cells. Three weeks post-transfection, colonies were either imaged under blue light for *GFP* expression or fixed in 4% paraformaldehyde and stained with methylene blue in PBS for the colony counting. Colonies used for FACS analysis (Fig. 4c) were trypsinized 22 days post-transfection to generate single-cell suspension and used for fluorescence microscopy. $1 \times 10^4$ cells were then assayed for the GFP fluorescence expression intensity (NHLBI Flow Cytometry Core). Cells were tested for mycoplasma contamination.

**DpnI replication assay**. One microgram of isolated LMW was digested overnight (~19 h) with 20U *DpnI* (NEB). One microliter of the *DpnI* digestion or of the undigested isolated LMW was electroporated to MegaX T1$^R$ DH10B electrocompetent *E. coli* cells (Thermo Fisher Scientific). In total, 70 and 140 white *Cam*$^+$ *E. coli* colonies arising from the electroporation of undigested and *DpnI*-digested LMW DNA, respectively, were analyzed by sequencing. Detailed description of the assay is provided in Supplementary Note 5.

**In vitro formation and detection of *Helraiser* end junctions**. *Helraiser* transposon junctions were generated using 0.3 µM purified transposase and 1 µg of linearized pHelR-Cam (~3.6 kb) plasmids or pHelR(mm)-Cam heteroduplex transposon donors in 30 µl total volume reactions. Reaction were performed for 60 min at 37 °C in 20 mM Hepes buffer of pH 7.3 containing 5% glycerol, 5 mM MgCl$_2$, 1 mM MnCl$_2$, 100 mM KCl, 1 mM ATP, 20 mM DTT, and 200 µM BSA. The reactions were stopped by adding 2 µl of 0.5 M EDTA and 2 µl proteinase K (NEB) followed by incubation at 37 °C for additional 30 min. Further details are provided in Supplementary Note 6.

**Protein expression and purification**. *Helraiser* transposase and point mutants were expressed in insect cells (service provided by GenScript) and proteins were purified from insect cell pellets as described previously[21].

**In vitro cleavage reactions**. In vitro cleavage reactions were performed using conditions described previously[21]. Oligonucleotides used in the cleavage assay are listed in the Supplementary table 1.

**Statistics and general methods**. The required sample sizes were estimated by considering variations and means, and sought to provide reliable conclusions. In the analysis of the sequenced transposon-end junctions unreadable sequencing results, sequences matching original transposon donors and those that lacked both transposon ends were excluded.

**Data availability**. All data used in the current research are available upon request to the corresponding author.

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

## Acknowledgements

We are grateful to Drs. Zoltan Ivics and Michael Chandler for critically reading the manuscript and for advice, Dr. Zoltan Ivics for encouragement and his ongoing interest

in the project, Dr. Qiujia Chen for useful advice and technical assistance with protein purifications, and Oliver Pavletic for his assistance in some of the experiments. This work was funded by the Intramural Research Program of the National Institute of Diabetes and Digestive and Kidney Diseases of the NIH.

## Author contributions

I.G. designed and performed the experimental work, interpreted the experimental results and wrote the paper. A.B.H. contributed to the interpretation of the experimental results and wrote the paper. F.D. provided direction, supervised the experimental work and wrote the paper.

## Additional information

**Competing interests:** The authors declare no competing interests.

