## [Peer Review File(PDF 305 kb) · Nature Communications]

Reviewers' comments:

Reviewer #1 (Remarks to the Author):

Helitrons are among the most ubiquitous and poorly understood of all eukaryotic transposons. As with many transposons, the vast majority of Helitrons are non-autonomous elements, many of which have transduplicated, or captured fragments of host genes. In maize, for instance, thousands of fragments of genes scattered throughout the genome. Until recently, our understanding of the means by which Helitrons propagate within eukaryotes was limited the lack of functional enzymes produced from an autonomous elements. Indeed, nearly everything we know about this rather unusual class of elements comes from in silico analysis and speculation based on limited similarity with a bacterial IS element. With their previous publication it would appear that this is no longer a limitation, which opens up a wide variety of experiments that can address important mechanistic questions. Given the novelty and ubiquity of these elements, these experiments represent an important step forward.

The authors are clearly experts in this kind of analysis. And this brings me to my only caveat: for anyone but a non-specialist, many of these experiments will be extremely difficult to follow. I would suggest that the author add a bit more explanatory text throughout the manuscript in order not to lose all but the most specialized readers. In that vein, I would also suggest that the authors seed the results with short explanatory sentences and phrases before discussing the results of each experiment, such as "if X was true, we would expect to see Y because of Z." As far as I can tell, there is nothing inaccurate with the results and the text is quite well written, but these minor changes could make the manuscript accessible to a wider audience.

The discussion is reasonable and highlights the novelty of the results. The idea that Helraiser DNA circles may act as episomes that can propagate independently is fascinating. I know that there is some research on something that is being called "circleomes"; one wonders how these result might inform that data.

Reviewer #2 (Remarks to the Author):

The experiments address an important and complicated question. Ideally, a full in vitro reaction system would be recapitulated to definitively unveil the mechanisms of transposition, but an impressive number of experiments are presented that are consistent with their model. These experiments are very clever but also very complicated. While they use a number of diagrams to try and make this clear, more effort is needed to make it broadly understandable across the transposon community and remotely approachable to people outside of transposition.

Major points

There should be more effort putting the work in context of the 2015. Too much is assumed and it is important to point out what was known from that work and what was made more clear with the current work.

Paper does not live up to the title of "unveiling the mechanism." A more nuanced title is needed to emphasize what was advanced specifically in this paper.

There should be mention that other reaction products may be unstable in the HEC293T cells. Therefore there could be biases.

Authors need to convincingly explain why the argument found in lines 325-329 is not circular. By their line of argumentation any reaction intermediate is important because it is a check point.

They should consider annealing the opposite polarity substrates they produced (i.e. ss+ and ss- in figure 2a and 2b) back to a dsDNA form to determine if they are now viable substrates for transposition. This would definitively rule out any alteration to the products that precluded transposition.

Other points...

Text referring to Figure 2B - To make it abundantly clear it should be explained that the circular transposon is a head-to-tail junction of the transposon ends (the 5'-TC dinucleotide of the LTS jointed to the CTAG-3') as described in the 2015 paper.

Line 74 - "rewire their host's genome regulatory networks" - jargon/grandiose, maybe diversity generating?

Line 82 - "unrelated ends" sounds like referring to phylogeny, maybe "distinct" or "very different types" of end structures?

Line 120 - Incorrect to say "High copy number" supports the notion of rolling circle transposition. Many types of transposons could lead to high copy number when unregulated.

Lines 125 - 134 - Use of "We" probably not justified in this paragraph given the extensive differences in the authors between the two documents.

Lines 125 - 134 - Comment on the cis-acting requirements of the end sequences and specific roles for the functional domains from the 2015 paper?

Line 230 - "dominantly," should be "predominantly"?

Numbering/order of the supplementary figures needs to be fixed.

Point-by-point responses to the Referees' comments:

Reviewer #1 (Remarks to the Author):

Helitrons are among the most ubiquitous and poorly understood of all eukaryotic transposons. As with many transposons, the vast majority of Helitrons are non-autonomous elements, many of which have transduplicated, or captured fragments of host genes. In maize, for instance, thousands of fragments of genes scattered throughout the genome. Until recently, our understanding of the means by which Helitrons propagate within eukaryotes was limited the lack of functional enzymes produced from an autonomous elements. Indeed, nearly everything we know about this rather unusual class of elements comes from in silico analysis and speculation based on limited similarity with a bacterial IS element. With their previous publication it would appear that this is no longer a limitation, which opens up a wide variety of experiments that can address important mechanistic questions. Given the novelty and ubiquity of these elements, these experiments represent an important step forward.

We thank the Referee for the succinct summary of the significance of our work and the kind evaluation.

The authors are clearly experts in this kind of analysis. And this brings me to my only caveat: for anyone but a non-specialist, many of these experiments will be extremely difficult to follow. I would suggest that the author add a bit more explanatory text throughout the manuscript in order not to lose all but the most specialized readers. In that vein, I would also suggest that the authors seed the results with short explanatory sentences and phrases before discussing the results of each experiment, such as “if X was true, we would expect to see Y because of Z.” As far as I can tell, there is nothing inaccurate with the results and the text is quite well written, but these minor changes could make the manuscript accessible to a wider audience.

We thank referee for this suggestion. We were acutely aware that some of the experiments are difficult to follow. As referee suggested, we have added a number of explanatory "guiding" text throughout the Results section.

The discussion is reasonable and highlights the novelty of the results. The idea that Helraiser DNA circles may act as episomes that can propagate independently is fascinating. I know that there is some research on something that is being called “circleomes”; one wonders how these result might inform that data.

Regrettably, we are not familiar with the term "circleome", and have been unable to locate any published work on the specific subject. However, we agree with the sentiment of the referee and it is clear that many interesting forms of circular DNA and RNA can exist in cells with a myriad of functions (including circular viral genomes, plasmids, and circRNA etc.). We feel it is premature to conclude anything about possible parallels at this stage of our work beyond the parallels to the replication of certain ssDNA viruses that we pointed out in Discussion.

Reviewer #2 (Remarks to the Author):

The experiments address an important and complicated question. Ideally, a full in vitro reaction system would be recapitulated to definitively unveil the mechanisms of transposition, but an impressive number of experiments are presented that are consistent with their model. These

experiments are very clever but also very complicated. While they use a number of diagrams to try and make this clear, more effort is needed to make it broadly understandable across the transposon community and remotely approachable to people outside of transposition.

Referee is correct, we are indeed in the process of attempting to recapitulate the various steps of *Helitron* transposition in vitro and also to determine what if any cellular factors are needed. As indicated for the comment written in the same vein from Reviewer #1, we have “seeded” many of the paragraphs in the Results section with guiding sentences and phrases to improve the overall accessibility of the paper.

Major points

There should be more effort putting the work in context of the 2015. Too much is assumed and it is important to point out what was known from that work and what was made more clear with the current work.

We have added a paragraph to the Introduction to summarize the results of the work published in 2016. We have also now attempted to point out in several places in the Results and the Discussion where the current work has provided novel insights.

Paper does not live up to the title of “unveiling the mechanism.” A more nuanced title is needed to emphasize what was advanced specifically in this paper.

In retrospect, the referee is correct that we are not quite at the point of having completely unveiled the mechanism. We have therefore replaced "unveil" with "provide insight into", which is perhaps less eye catching, but is more accurate.

There should be mention that other reaction products may be unstable in the HEC293T cells. Therefore there could be biases.

Referee is correct and this has now been noted on page 15.

Authors need to convincingly explain why the argument found in lines 325-329 is not circular. By their line of argumentation any reaction intermediate is important because it is a check point.

We agree that, as written, the argument was unclear. The sentence has been revised to read: "One possible interpretation of this result is that dsDNA circles containing a precise RTS-to-LTS junction are preferred substrates for subsequent *Helraiser* circle replication."

They should consider annealing the opposite polarity substrates they produced (i.e. ss+ and ss- in figure 2a and 2b) back to a dsDNA form to determine if they are now viable substrates for transposition. This would definitively rule out any alteration to the products that precluded transposition.

We agree that such a control would be desirable. However, it has been shown that hybridization of two complementary circular ssDNA molecules results in the formation of a ds circular DNA molecule, form V, which has properties similar to that of left-handed ds Z-DNA (Stettler et al. *J. Mol. Biol.* **131** 21-40 (1979); Wang et al., *Nature* **282**, 680 (1979)). Since form V DNA is both relatively unstable (Stettler UH *et al.*) and adopts an unusual conformation, it is not clear if it would be recognized by the *Helraiser* transposase, even if it was possible to successfully transfect it into cells.

Of course, one way to circumvent the problem of form V DNA would be to convert it to B-form DNA using topoisomerase. During the early stages of our work, we attempted exactly this but, despite considerable efforts, in our hands this resulted in the formation of multiple DNA species and we were not able to isolate the desired product in sufficient quantities for subsequent experiments.

We have therefore performed a close approximation to the experiment suggested by the referee in which we have annealed the ssDNA to the linearized form of the plasmid that was initially used to generate circular ssDNA as outlined in the accompanying figure (a).

Characterization of ssDNA transposon donors (a) Generation of heteroduplex transposon donors using ssDNA and linearized plasmid DNA. DNA sequences of pHeIR-CM1 and pHeIR-CM1 (A/T) differ only in one basepair. Relevant DNA substrates are in capital letters. This annotation is used consistently in the figure. (b) Circle and donor detection PCRs spanning the mismatch region. LMW DNA isolated post transfection with heteroduplex transposon donors D and E was used in PCRs. M, marker. (c) Circle detection PCR using LMW DNA isolated post transfection with heteroduplex donors E and pHeIR-CM1 (A). (d) S1 nuclease digestion of ssDNA transposon donors (B and C). (e) Exonuclease I digestion of the ssDNA transposon donors C.

To ensure that the obtained relaxed circular dsDNA was indeed generated in the described process, we used linearized plasmids that differed in one bp from the plasmid used to generate circular ssDNA. Thus, the desired products should contain a single mismatch, distinguishing them from their respective starting molecules. We confirmed by DNA sequencing that the generated molecules were indeed in heteroduplex form and that they maintained this state post-transfection and LMW isolation. When we used these heteroduplex transposon donors to test for *Helraiser* circle formation in HEK293T cells, we found that they were efficiently used and yielded circular transposon intermediates with the same efficiency as authentic donors (**b**, **c**). Thus, although the dsDNA donors we produced are not identical to the control suggested by the referee, each version contained one strand originating from the circular ssDNA. Since we generated and tested heteroduplexes made from both strands of circular ssDNA, we believe this addresses the referee's concern to a large extent. We have decided at this point not to include this experiment in the paper as it is already long, however we can certainly include it in Supplementary Materials should that be needed.

The other controls worth mentioning are that we attempted to rule out any obvious alterations in produced circular ssDNA by treating generated ssDNA with S1 endonuclease (**d**; digests ssDNA) and Exonuclease I (**e**; digests linear but not circular ssDNA). Again, these could also be appended as Supplementary material, if needed.

Other points...

Text referring to Figure 2B - To make it abundantly clear it should be explained that the circular transposon is a head-to-tail junction of the transposon ends (the 5'-TC dinucleotide of the LTS jointed to the CTAG-3') as described in the 2015 paper.

This explanation (precisely as written by the referee) has been added to the first reference to the circular transposon on page 6.

Line 74 - "rewire their host's genome regulatory networks" - jargon/grandiose, maybe diversity generating?

The phrase has been rewritten as "impact the functional organization of their host's genome regulatory networks".

Line 82 - "unrelated ends" sounds like referring to phylogeny, maybe "distinct" or "very different types" of end structures?

As suggested, the phrase has been rewritten as "each end contains a distinct ~150 basepairs (bp) long sequence".

Line 120 - Incorrect to say "High copy number" supports the notion of rolling circle transposition. Many types of transposons could lead to high copy number when unregulated.

This is a good point, and the referee is indeed correct that there is not a logical link between high transposon copy number and a particular mode of transposition. We have removed both references to high copy numbers in this paragraph.

Lines 125 - 134 - Use of "We" probably not justified in this paragraph given the extensive differences in the authors between the two documents.

We agree, and have rewritten the paragraph to more accurately represent the contributions of the various authors by referring instead to "Grabundzija et al." and "this study", rather than "we".

Lines 125 - 134 - Comment on the cis-acting requirements of the end sequences and specific roles for the functional domains from the 2015 paper?

As requested by the referee, we have added several sentences outlining the results obtained from the 2016 paper. These are highlighted in yellow on page 7.

Line 230 - "dominantly," should be "predominantly"?

Thank you. This has been corrected.

Numbering/order of the supplementary figures needs to be fixed.

They have been corrected.

REVIEWERS' COMMENTS:

Reviewer #2 (Remarks to the Author):

I am very satisfied with the new version of the manuscript. The concerns I had were addressed experimentally and with the rewording of the manuscript.

One small suggestion is at the end of line 118 to include "...at staggered positions which are subsequently repaired by a gap-filling mechanism." (joining to both DNA stands will only make a TSD if they are at staggered positions.)

Point-by-point responses to the Referees' comments:

REVIEWERS' COMMENTS:

Reviewer #2 (Remarks to the Author):

I am very satisfied with the new version of the manuscript. The concerns I had were addressed experimentally and with the rewording of the manuscript.

One small suggestion is at the end of line 118 to include "...at staggered positions which are subsequently repaired by a gap-filling mechanism." (joining to both DNA stands will only make a TSD if they are at staggered positions.)

The reviewer is correct that the sentence as written was inaccurate, and has been revised as suggested. We would like to thank the reviewer for the careful reading of our manuscript and the very helpful suggestions for improvements.